# The Impact of Diabetes on Outcomes for Tibiotalocalcaneal Arthrodesis: A Systematic Review of Available Comparative Studies

**DOI:** 10.3390/healthcare13040385

**Published:** 2025-02-11

**Authors:** Grayson M. Talaski, Anthony N. Baumann, Nicholas I. Chiaramonti, Nolan M. Schonhorst, Conor N. O’Neill, Kempland C. Walley, Albert T. Anastasio, Samuel B. Adams

**Affiliations:** 1Department of Orthopedics and Rehabilitation, University of Iowa, Iowa City, IA 52242, USA; nmschonhorst@uiowa.edu; 2College of Medicine, Northeast Ohio Medical University, Rootstown, OH 44272, USA; abaumann@neomed.edu; 3College of Medicine, Central Michigan University, Mount Pleasant, MI 48859, USA; chiar1ni@cmich.edu; 4Department of Orthopaedic Surgery, Duke University, Durham, NC 27708, USA; conor.n.oneill@duke.edu (C.N.O.); albert.anastasio@duke.edu (A.T.A.); samuel.adams@duke.edu (S.B.A.); 5Department of Orthopaedic Surgery, University of Michigan, Ann Arbor, MI 48109, USA; kcwalley@med.umich.edu

**Keywords:** tibiotalocalcaneal arthrodesis, diabetes mellitus, systematic review

## Abstract

**Background/Objectives:** Tibiotalocalcaneal (TTC) arthrodesis is commonly used in salvage situations involving the ankle and subtalar joint, often in patients with concomitant diabetes mellitus (DM). Across orthopedics, DM presents an overall increased risk of developing complications post-surgically. In this systematic review, the primary aim was to summarize the outcomes and complications of patients undergoing TTC arthrodesis with DM. **Methods:** A qualitative systematic review was conducted, with an initial search performed on 30 August 2023, using PubMed, SPORTDiscus, CINAHL, and MEDLINE. The search algorithm “tibiotalocalcaneal” AND (nail OR nails) AND (fusion OR arthrodesis) was applied, following PRISMA guidelines. Inclusion criteria encompassed articles examining the impact of diabetes on TTC arthrodesis outcomes. Data extraction involved patient demographics, complication rates, and surgical outcomes. Due to data heterogeneity, a narrative approach was utilized to describe results across studies. **Results:** Four articles met the inclusion criteria. These observational comparative studies were of moderate quality, with a mean MINORS score of 20.5 ± 1.9 points. The combined patient cohort included 162 patients, evenly split between diabetic and non-diabetic groups, with a mean age of 58.2 ± 2.7 years and a follow-up duration of 35.0 ± 7.4 months. Diabetic patients exhibited higher rates of superficial infection, though functional outcomes and fusion rates were generally favorable. **Conclusions:** TTC arthrodesis in diabetic patients was associated with an increased risk of superficial infections and various other complications. Despite these risks, functional outcomes and rates of successful fusion were comparable to non-diabetic patients undergoing TTC arthrodesis. This review highlights the need for standardized definitions of surgical success.

## 1. Introduction

Tibiotalocalcaneal (TTC) arthrodesis is applied as a salvage procedure for end-stage conditions involving the ankle and subtalar joints, including unstable ankle or pilon fractures in low-demand patients, Charcot neuropathic joint destruction, and complex deformities involving both peritalar and subtalar instability [1,2]. In the setting of acute trauma, TTC is often used for cases when open reduction or internal fixation is not viable or would be excessively risky with regard to wound-related complications, offering a suitable alternative to lower limb amputation for many patients [3,4]. Overall outcomes for TTC are reported as positive, with one systematic review reporting nearly 82% of patients being able to return to pre-injury activity levels [5]. TTC arthrodesis requires less soft-tissue dissection compared to alternative procedures, and in addition to the mechanical advantages that an intermedullary nail brings in terms of stability, many complicated ankle conditions may be optimally treated with this procedure [3,6,7].

In recent years, diabetes mellitus (DM)-related neuropathic ankle disorders and acute ankle trauma in patients with poorly controlled DM have become increasingly common injury patterns, with numerous studies attributing this trend to the worldwide aging population and the sharp rise in the incidence of metabolic syndrome [8,9,10,11]. As demonstrated widely across the literature, patients with DM are at higher risk of developing postoperative complications, regardless of the procedure [12,13,14]. Thus, the primary goal of TTC arthrodesis in diabetic patients often shifts away from complete deformity correction, and instead towards limb salvage and ambulatory preservation [14].

The scientific literature describing outcomes of TTC arthrodesis in DM patient populations has grown in recent years; however, a comprehensive systematic review on this topic has yet to be performed. As the complexity of diabetic TTC arthrodesis cases can be heterogenous, a systematic review is recommended in order to better describe the outcomes of this procedure in a DM cohort. Therefore, the objective of this systematic review was to summarize findings pertaining to TTC arthrodesis in the DM population, with the goal of understanding its outcomes and complications.

## 2. Materials and Methods

### 2.1. Study Creation

This study is a qualitative systematic review, with an initial search conducted on 30 August 2023, using PubMed, SPORTDiscus, CINAHL, and MEDLINE. These four databases were searched from database inception until the search date, and the search algorithm used was “tibiotalocalcaneal” AND (nail OR nails) AND (fusion OR arthrodesis), using the most recent Preferred Reporting Items for Systematic Reviews and Meta-Analyses (PRISMA) guidelines [15]. This study’s protocol was not pre-registered on PROSPERO.

### 2.2. Inclusion and Exclusion Criteria

The inclusion criteria were articles that examined the impact of DM on any outcome after TTC arthrodesis, randomized controlled trials (RCTs), and comparative, case–control studies. Exclusion criteria were articles that did not stratify by DM status, used TTC arthrodesis fixation constructs other than the intramedullary nail (IMN), case reports, systematic reviews, meta-analyses, non-English articles, and those with a lack of a full text. Articles with overlapping patient cohorts were examined, and only the more recent study was included.

### 2.3. Article Sorting Process

Article sorting was completed by a single author. After the search was performed, the articles were downloaded into Rayyan for ease of article sorting [16]. Duplicate articles were removed, followed by screening of titles and abstracts, and, finally, full-text screening.

### 2.4. Data Extraction Process

Data extraction was completed by multiple authors. Articles were screened by two authors (N.I.C. and N.M.S.), with disagreements being solved by the senior author. The data extracted included the first author, year of publication, treated condition, method of fixation, number of patients, group descriptions, age, male/female status, body mass index, follow-up time, patient comorbidities, injury type, complication rates, successful arthrodesis rates, pseudoarthrosis rates, limb salvage rates, and amputation rates.

### 2.5. Article Quality Grading

The Methodological Index for Non-Randomized Studies (MINORS) scale was used to assess each included study [17]. For comparative studies, the MINORS scale has 12 items, each worth 0–2 points, for a total of 24 points. Based on precedents in the literature [18], studies with 24 points were designated as being of “high quality”, whereas studies with 15–23 points or <15 points were designated as being of “moderate quality” and “low quality”, respectively.

### 2.6. Statistical Considerations

The Statistical Package for the Social Sciences (SPSS) version 29.0 (Armonk, NY, USA: IBM Corp) was used for statistical analysis. Due to the relatively small number of articles and heterogenous outcomes, a qualitative and narrative approach was taken. Frequency weighted means (FWMs), along with other descriptive statistics, were used to describe the data.

## 3. Results

### 3.1. Initial Search and Article Grading Results

A total of four articles met the inclusion criteria out of the 677 articles yielded from the search terms (Figure 1; PRISMA diagram) [19,20,21,22]. All four articles were observational comparative studies, and determined to be of “moderate quality” (Table 1; MINORS results). The mean MINORS score (n = 4 articles) was 20.5 ± 1.9 points (minimum–maximum: 18.0–22.0 points). No RCTs were found on this topic, indicating a lack of high-level evidence on this topic.

### 3.2. Patient Demographics

Patients (n = 162; 46.9% male) had an FWM age of 58.2 ± 2.7 years, with an FWM follow-up of 35.0 ± 7.4 months (Table 2; demographic results). All patients underwent TTC arthrodesis with IMN, and were divided based on diabetic status, with 81 patients (50.0%) having diabetes and 81 patients (50.0%) not having diabetes. More information regarding patient demographics is listed in Table 2.

### 3.3. Intervention Indications

The most common indication for TTC arthrodesis in the diabetic subgroup was Charcot arthropathy, with 65% of the pooled patient cohort listing Charcot as the reason for intervention. This was higher than in the non-DM subgroup, where Charcot was listed as the primary indication in only 12.6% of patients. Both groups had similar proportions of arthritic patients and revision-related interventions. In the study by Wukich et al., both subgroups were similar in age, gender, length of surgery, BMI, tobacco use, previous surgery, previous foot ulcer, and rheumatoid arthritis status [22]. However, the DM patient subgroup had significantly higher serum glucose, creatinine, and hemoglobin A1c (HbA1c) levels. Additionally, DM patients had lower preoperative hemoglobin levels. DM patients were also more likely to have peripheral neuropathy, Charcot neuroarthropathy (CN), and arterial disease as comorbidities [22].

Regarding intervention indications across individual studies, Lee et al. listed intervention indications as ankle and subtalar arthritis (50%, n = 10), Charcot arthropathy (35%, n =7), and equinovarus deformity (15%, n = 3) [19]. Bibbo et al. listed indications as Charcot ankle and hindfoot (40%, n = 2), ankle and subtalar arthritis (60%, n = 3), and avascular necrosis (AVN) of the talus (20%, n = 1) [20]. Mendicino listed common indications as post-traumatic arthritis, AVN of the talus, rheumatoid arthritis, Charcot neuroarthropathy, and severe deformity secondary to clubfoot or neuromuscular disease [21]. Diabetic patients in Wukich et al.’s study displayed intervention indications of Charcot neuroarthropathy (72.13%, n = 44), subtalar and ankle arthritis (13.12%, n = 8), post-traumatic arthritis (9.84%, n = 6), failed total ankle replacement (1.64%, n = 1), and revision arthrodesis (3.28%, n = 2) [22].

### 3.4. Follow-Up Duration

In the study by Wukich et al., the mean follow-up period was 159.8 ± 92.9 weeks for patients with DM, and 38.78 ± 22.45 weeks for patients without DM [22]. Mendicino et al. reported an average follow-up of 19.8 months (range: 8–42) [21]. Similarly, Bibbo et al. reported a mean follow-up of 20.2 months [20]. Lee et al. extended the follow-up period to a mean of 28.6 months (13–49) [19].

### 3.5. Complication Rates

Across studies, complications were often classified as major, minor, infectious, or noninfectious. Wukich et al. and Lee et al. classified infectious complications as superficial wound infections and deep infections [19,22], whereas Wukich et al. classified noninfectious complications as nonunion, symptomatic hardware removal, and postoperative tibia fractures [22]. Mendicino et al. classified major complications as osteomyelitis, acute Charcot nonunion, continued Charcot process, loss of fixation, full-thickness necrosis, and major medical complications [21]. Minor complications comprised ulcers 2° to the nail, mild wound dehiscence, limb length discrepancies, and chronic wound drainage [21]. Lee et al. classified major complications as implant failure, implant removal, revision surgery, and amputation, whereas minor complications comprised postoperative blood transfusion and decreased hemoglobin levels [19]. Bibbo et al. listed major complications as amputation and implant removal/failure and infectious-based complications [20].

Regarding infection-based complications, Wukich et al. indicated that patients with DM were eight times more likely to demonstrate superficial infection (*p* = 0.03) [22]. Similarly, Mendicino et al. found that DM patients experienced higher rates of medical and surgical complications, with a major complication rate of 50% [21]. Bibbo et al. described infectious complications in all five patients, with infections including Staphylococcus epidermis (n =3), methicillin resistance (n = 1), Corynebacterium jeikeium (JK) infection (n = 1), enterococcus species infection (n = 2), and prevotella buccae infection (n = 1) [20]. All organisms were sensitive to the antibiotics used in the antibiotic nails [20].

Regarding noninfectious complications, Wukich et al. found no significant difference between patients with and without DM (*p* = 0.0877), with 24.6% of diabetic patients experiencing noninfectious complications, compared to 39.3% of non-DM patients [22]. In Mendicino et al.’s study, major noninfectious complications in diabetic patients included osteomyelitis and progression of Charcot arthropathy [21]. Bibbo et al. did not specifically detail noninfectious complications [20], while Lee et al. reported a postoperative hematoma in one case, requiring surgical intervention [19].

### 3.6. Functional Outcomes

Functional outcomes were generally favorable across the studies. Wukich et al. reported that more than 95% of patients in both groups were ambulatory at the most recent follow-up (*p* = 0.60), with a trend towards greater brace use in DM patients (*p* = 0.0569). In Mendicino et al.’s study, 17 out of 19 patients ambulated without assistive devices at final follow-up [21]. Bibbo et al. documented clinical freedom from infection, with stable pseudarthrosis allowing household ambulation [20]. Lee et al. found statistically significant improvements in the Short-Form 36-Item Questionnaire (SF-36) and American Orthopaedic Foot and Ankle Society Ankle Hindfoot Scale (AOFAS AHS), with 95% of patients reporting favorable outcomes and willingness to recommend the surgery.

### 3.7. Fusion Rates

Fusion/union rates were highly successful across studies. Mendicino et al. reported a 95% fusion rate at both the ankle and subtalar joints, with fusion defined by osseous bridging consistent with consolidation measured via radiographs [21]. Lee et al. found no significant difference in fusion rates between subgroups, with fusion being defined as a fusion mass of grade ≥ 2 (grade 1, 0% to 25% of the joint surface; grade 2, 26% to 50% of the joint surface; grade 3, 51% to 75% of the joint surface; and grade 4, 76% to 100% of the joint surface) according to the most recent anteroposterior radiographs [19]. Bibbo et al. were not able to comment on statistical differences between subgroups due to a small cohort, but fusion was measured radiographically, with many patients requiring refusion intervention [20]. Finally, Wukich et al. reported similar fusion rates between DM (83.6%) and non-DM (75%) subgroups. Fusion was classified as more than 50% osseous fusion, measured via radiographs or computed topography scans [22].

### 3.8. Use of Biologics and Bone Grafts

To encourage fusion, Mendicino et al. and Lee et al. did not utilize any biologics, and relied on cancellous, autogenous bone grafts from the distal fibula to fill bony defects [19,21]. Wukich et al. utilized femoral head allografts in 27.4% of patients, but did not denote the frequency of this graft in DM and non-DM subgroups [22]. In this study, patients with femoral head allografts had a higher rate of complications than patients that received no femoral head allograft, with rates of 67.7% and 37.6% (*p* = 0.008) [22]. While the use of femoral head allografts was necessary in selected patients, particularly due to talar body deficiencies, the overall risk of complication was higher [22]. The study did not differentiate which type of complication was more prevalent for these patients, but overall complications comprised both infectious and noninfectious complications [22].

### 3.9. Postoperative Concerns

Three out of the four of the included articles reported on surgical complication rates with a total complication rate of 47.7% (n = 75 cases out of 157 total patients) for the entire patient population (Table 3; complications and postoperative concerns results). Two of these articles stratified complication rate by diabetic status; overall, surgical complication rates were similar, with DM patients having a surgical complication rate of 47.9% (n = 34 cases out of 71 patients) and non-DM patients having a surgical complication rate of 50.0% (n = 33 cases out of 66 patients).

## 4. Discussion

In this systematic review, TTC arthrodesis with IMN in diabetic patients was evaluated across four comparative studies. As DM patients often suffer from peripheral neuropathy, with nearly 40% of patients developing this comorbidity within one decade [23,24], understanding DM risks is crucial in order to mitigate the risk of failed intervention. While TTC is a relatively successful procedure [3], the risk of amputation is increased due to neuropathy and ulceration in DM patients. According to one of the largest diabetic limb registries, 52% of patients who received a below-knee amputation were deceased within 2 years [25]. Furthermore, this registry found that DM patients fear the loss of a limb more than death and even end-stage renal disease [25]. Therefore, foot and ankle orthopedic surgeons hold great responsibility to not only successfully salvage end-stage ankle and subtalar joint conditions, but also to provide life-changing care for DM patients.

To date, no review has described the outcomes of TTC arthrodesis in DM patients compared to non-diabetic patients. As the case difficulty and health status of DM patients ranges across patient samples, the results of individual studies may not reflect the risks of performing TTC arthrodesis in diabetic populations. Therefore, this review aims to provide the most comprehensive analysis of complications, risks, and recommendations for TTC arthrodesis in diabetic patients.

Regarding infection-based complications, DM patients demonstrated higher complication rates. Specifically, Wukich et al. states that DM patients were 8 times more likely to develop superficial wound infections [22]. As superficial infections can prove difficult to heal in patients with DM [26], these findings reiterate the importance of proper wound healing and a careful surgical approach. Despite this finding, orthopedic surgeons have, overall, achieved a relatively low superficial wound infection rate for TTC arthrodesis of 10%. This number is particularly reassuring when considering the high rates of complications associated with nonoperative treatment of serious injuries to the ankle joint. For example, treatment of unstable ankle fractures with nonoperative treatment reveals major complication rates as high as 75%, with many complications resulting in amputation [12].

Union rates were favorable across both subgroups. Stable, radiographic union was reported at rates well within the acceptable values stated in the literature for this procedure in the DM cohort [27]. Functional outcomes were similar across both subgroups, with only one study reporting a higher usage of ambulatory devices in diabetic patients [22]. Moreover, functional outcome scores improved for both the DM subgroup and the non-DM cohorts. It is important to note that while DM patients achieved similar functional outcomes, the poor preoperative health status of included non-diabetic patients could confound this finding [21]. Varying study-specific definitions of operative success presents a discrepancy in results that must be addressed. While some studies defined success as stable union [21], some as limb preservation [20,22], and one study as a mix of both [19], it is important to establish a clear definition of success that can be applied across clinical settings. Even common metrics, such as nonunion, were not reported consistently across studies. Wukich et al. defined nonunion as less than 50% osseous bridging on radiographs or CT after 12 months [22], Lee et al. defined nonunion according to a previously published grading scale [19,28], and Mendicino (2004) and Bibbo (2003) did not define nonunion [20,21]. However, it is important to note that the definitions of nonunion have improved over time, and the heterogeneity across the included studies is likely an example of this.

As proper union is crucial for the long-term health of DM patients [29], preparation of the articular surface is crucial in order to ensure proper fusion. Successful results were demonstrated in the study by Mendicino et al. when approximately 5 to 10 mm of bone was removed from the lateral aspect of the medial malleolus and the medial aspect of the talus [21]. Furthermore, usage of distal fibulectomy provided autogenous bone grafts in successful cases across two studies [19,21]. Wukich described the use of femoral head allografts, but did not differentiate DM and non-DM subgroups. However, results for the femoral head allograft were less favorable [22]. While DM patients were not accessed directly in this study [22], the poor femoral head allograft results agree with a study performed by Jeng et al. [30]. This study found a nearly 50% nonunion rate for their entire cohort, and a 100% nonunion rate for DM patients when utilizing femoral head allografts [30]. While this study did not meet this review’s inclusion criteria, the results are worth noting, nonetheless.

While the included articles described joint preparation, numerous studies have advocated for the avoidance of joint preparation in complicated TTC cases [31,32]. Several authors argue that the absence of motion, combined with cartilage compression, is sufficient to simulate fusion [32]. In elderly patients with multiple comorbidities, a lengthy correction procedure to properly prep the subtalar and tibiotalar joints may not be worth the added risk [32]. As additional soft tissue dissection is required for proper debridement of the ankle joints, immediate weightbearing becomes difficult [32]. Early weightbearing is crucial for TTC with IMN, as reduced mobilization can increase the likelihood of complications and mortality [33]. In general, TTC arthrodesis leads to quicker mobilization compared to open reduction or internal fixation, due to the mechanical stiffness of the construct serving as an “internal brace” [32].

It is important to address the limitations of this study. First, data collection and heterogeneity in the definition of success prevented meta-analysis in this review. Future research should focus on the development of standardized results metrics to ensure robust comparison across institutions. Second, this study strictly compared DM patients to non-diabetic patients. Limited studies have analyzed TTC arthrodesis results in patients with varying DM severity [34], and a dedicated systematic review of this topic would greatly contribute to a comprehensive analysis of this procedure in DM patient populations. While diabetic severity, grouped as either “controlled” or “uncontrolled” diabetes in previous studies, did demonstrate higher TTC failure rates [34], additional factors such as age, BMI, and length of diabetes diagnosis all may impact the success of this procedure. Therefore, there is a need for detailed future studies that help to provide an understanding of the outcomes in DM patients, while accounting for the previously mentioned confounding factors. Finally, all included articles were retrospective, comparative studies. Retrospective studies come with numerous inevitable biases that call for future prospective studies. However, retrospective studies remain the only comparative articles that can be included in a robust systematic review at this time. Nonetheless, this review adds great value to the literature by combining articles with varying patient preoperative health statuses, case complexities, and cohort sizes.

## 5. Conclusions

In conclusion, this review evaluated TTC arthrodesis outcomes in DM patients. The primary concern for DM patients was found to be superficial wound infection, with a nearly 8-fold greater likelihood. Postoperative success, including radiographic union and functional outcomes, was comparable between DM and non-diabetic patients. The review’s limitations include data collection heterogeneity, the inability to perform a meta-analysis, and reliance on retrospective studies.

## Figures and Tables

**Figure 1 healthcare-13-00385-f001:**
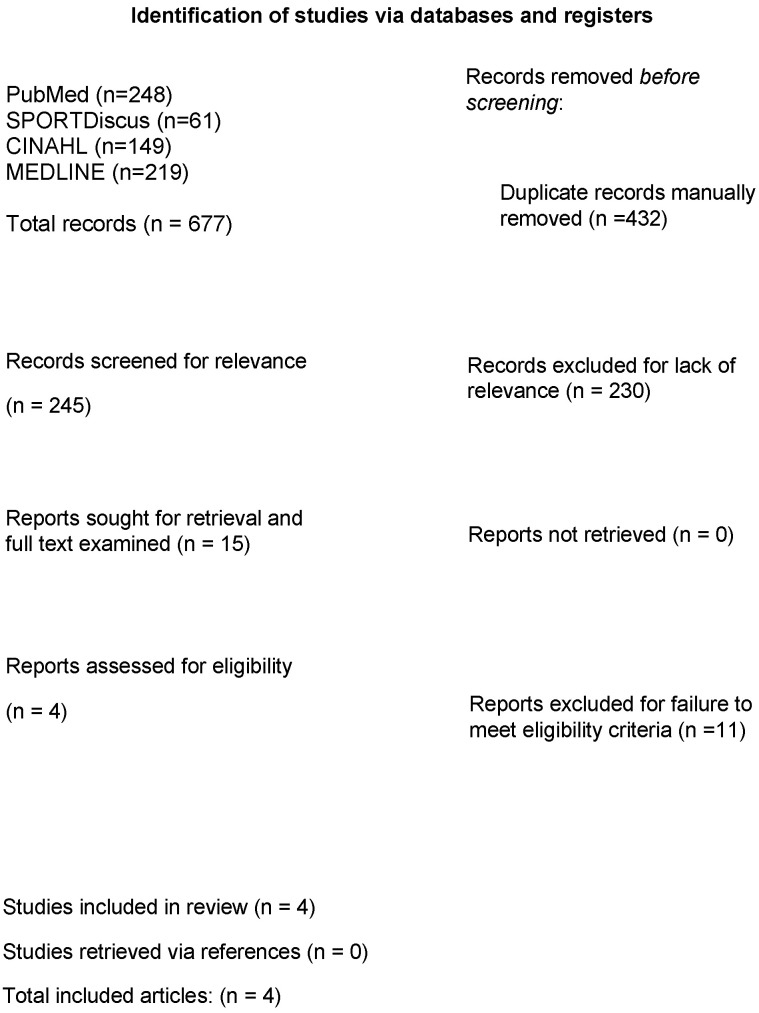
Preferred Reporting Items for Systematic Reviews and Meta-Analyses (PRISMA) diagram.

**Table 1 healthcare-13-00385-t001:** The Methodological Index for Non-Randomized Studies (MINORS) results for each study.

Author (Year)	Study Type	Total MINORS Score	Clearly Stated Aim	Inclusion of Consecutive Patients	Prospective Collection of Data	End Points Appropriate to Study Aim	Unbiased Assessment of Study End Point	Follow-Up Period Appropriate to Study Aim	Less Than 5% Lost to Follow-Up	Prospective Calculation of Study Size	Adequate Control Group	Contemporary Groups	Baseline Equivalence of Groups	Adequate Statistical Analysis
Lee (2017) [19]	Comparative	22	2	2	2	2	2	2	2	0	2	2	2	2
Bibbo (2003) [20]	Comparative	20	2	2	2	2	2	2	2	0	2	2	2	0
Mendicino (2004) [21]	Comparative	18	2	0	0	2	2	2	2	0	2	2	2	2
Wukich (2015) [22]	Comparative	22	2	2	2	2	2	2	2	0	2	2	2	2

MINORS—Methodological Index for Non-Randomized Scale.

**Table 2 healthcare-13-00385-t002:** Demographic information from the included articles in this systematic review.

Author (Year)	Treated Condition	Method of Fixation	Group Size	Group Description	Age (Years) Mean (Range/SD) or Median [IQR]	Gender Info	BMI (Mean (Range/SD) or Median [IQR])	Months of Clinical Follow-Up (Mean (Range/SD))	Comorbidities (%)	Injury Type (n, %)
Lee (2017) [19]	Ankle and subtalar arthrosis, previous failed TTCA with other implants, and severe hindfoot deformities that were refractory to other treatment, such as Charcot arthropathy and severe equinovarus deformity	TTCA	20	All patients	61.1 (39–78)	Male (N = 8, 40%), female (n = 12, 60%)		28.6 (13–49)	Smoker (3, 15%)	Joint arthritis (10, 50%), Charcot arthropathy (7, 35%), equinovarus deformity (3, 15%)
			7	DM subgroup	60.0 (46–78)	Male (n = 2, 28.6%), female (n = 5, 71.4%)		26.7 (13–40)	Poor glycemic control (2, 10%)	Charcot arthropathy (85.7%)
Bibbo (2003) [20]	Infected TTCA-RIMN	PMMA Nail with antibiotics	2	No diabetes	41 (24–58)	Male (n = 2, 100%)		20.2	Smoker (100%)	High-energy pylor fractures (1 open, 1 closed)
			3	Diabetes	64 (54–70)	Male (n = 2, 67), female (n = 1, 33%)			Diabetes (100%), neuropathy (100%), peripheral vascular disease (67%), cardiopulmonary disease (67%)	Charcot arthropathy with deformity, or avascular necrosis of talus with ankle/subtalar joint arthritis
Mendicino (2004) [21]	Various hindfoot pathologies	TTC	10	No diabetes	56 (33–81)	Male (n = 8, 42.1%), female (n = 11, 57.9%)		19.8 (8–42)	Diabetic neuropathy (n = 7), smoker (n = 5), primary osteoarthritis (n = 4), post-traumatic arthritis (n = 2), rheumatoid arthritis (n = 2), gouty arthritis (n = 2), equinocavovarus (n = 2), ankle malunion (n = 1)	
			10	Diabetes						
Wukich (2015) [22]	Arthritis of ankle and foot, deformity of ankle or foot	TTC	56	No diabetes	56.9 (12.8)	Male (n = 25, 45.5%), female (n = 31, 55.4%)	32.3 (6.01)	38.775 (22.45)	Peripheral neuropathy (27, 48.2%), peripheral arterial disease (1, 1.8%), history of foot ulcer (14, 25.0%), renal disease (2, 3.6%), rheumatoid arthritis (4, 7.1%)	Charcot neuroarthropathy (7, 12.5%), osteoarthritis of ankle and subtalar joint (10, 17.86%), traumatic arthritis (12, 21.43%), acquired equinovarus deformity (9, 16.07%), failed total ankle replacement (2, 3.57%), revision arthrodesis (11, 19.64%), avascular necrosis of talus (5, 8.93%)
			61	Diabetes	59.4 (12.3)	Male (n = 31, 50.8%), female (n = 30, 49.2%)	32.7 (8.0)	39.95 (23.225)	DM type 1 (12, 19.7%), DM type 2 (49, 80.3%), peripheral neuropathy (58, 95.1%), peripheral arterial disease (12, 19.7%), history of foot ulcer (24, 39.3%), renal disease (21, 34.4%), rheumatoid arthritis (3, 4.9%)	Charcot neuroarthropathy (44, 72.13%), osteoarthritis of ankle and subtalar joint (8, 13.12%), traumatic arthritis (6, 9.84%), failed total ankle replacement (1, 1.64%), revision arthrodesis (2, 3.28%)

SD—standard deviation, IQR—interquartile range, n—number of patients.

**Table 3 healthcare-13-00385-t003:** Complication, arthrodesis/pseudoarthrosis, limb salvage, and amputation rates for each study.

Author (Year)	Surgical Complication Rate (n, % (Cause))	Successful Arthrodesis (n, %)	Stable Pseudoarthrosis (n, %)	Limb Salvage Rate (n, %)	Amputation Rate (n, %)
Lee (2017) [19]	8, 40% (7 infection, 1 postoperative hematoma at lateral malleolus)	20, 100%	-	-	-
Bibbo (2003) [20]	-	-	-	2, 100%	-
	-	-	2, 67%	3, 100%	-
Mendicino (2004) [21]	6, 60% (0 major, 6 minor)	10, 100%	-	-	-
	8, 80% (5 major, 5 minor)	9, 90%	-	-	-
Wukich (2015) [22]	27 (48.2%)	-	-	-	3 (5.4%)
	26 (42.6%)	-	-	-	2 (3.3%)

n—number of patients.

## Data Availability

The original contributions presented in this study are included in the article. Further inquiries can be directed to the corresponding author(s).

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
