# Peer review of "The Impact of Diabetes on Outcomes for Tibiotalocalcaneal Arthrodesis: A Systematic Review of Available Comparative Studies"

_healthcare, 2025, doi:10.3390/healthcare13040385_

Round 1

Reviewer 1 Report

Comments and Suggestions for Authors

Thank you for the privilege of reviewing your manuscript: The impact of Diabetes on Outcomes for Tibiocalcaneal Arthrodesis: A Systematic Review of Available Comparative. I read your article with interest.

The topic falls within the scope of healthcare.

Authors showed that the predominant risk of patients with diabetes mellitus undergoing tibiocalcaneal arthrodesis is superficial wound infection (8-fold greater likelihood). However, authors noted that post-operative success, including radiographic bone-union and functional outcomes, were comparable between DM and non-diabetic patients.

Please correct some words and phrases:

Line 116: replace “pylor fractures” by “pilon fractures”.

Please have a close look at the correct citation of references. I think in some references more names of co-authors need to be shown.

Author Response

Comment 1: Line 116: replace “pylor fractures” by “pilon fractures”.

Response 1: Thank you for catching this wording mistake. I have corrected this mistake.

Reviewer 2 Report

Comments and Suggestions for Authors

This manuscript addresses an important issue in orthopedic surgery: the impact of diabetes on TTC arthrodesis outcomes. Given the increasing prevalence of diabetes and its complications, this review is relevant and valuable. The authors highlight the lack of RCTs and call for standardized definitions of surgical success.

Comments:

1.      It is surprising that only 4 studies met the inclusion criteria out of 677 initially identified articles (indicates bias in study selection in addition to it was performed by only one person). This small sample size could the generalizability of the findings.

2.      It would be adding more value to have a forest plot to visualize some of the findings.

3.      A more nuanced discussion on diabetes severity and its specific impact on outcomes could add more depth.

4.      The authors conclude that diabetic and non-diabetic patients have comparable functional outcomes, but can this be misleading? – given only few studies reviewed.

5.      The study fails to establish a clear, standardized definition of success across studies. Is it functional measures? Or fusion rate?

Author Response

Reviewer 2:

Comment 1: It is surprising that only 4 studies met the inclusion criteria out of 677 initially identified articles (indicates bias in study selection in addition to it was performed by only one person). This small sample size could the generalizability of the findings.

Response 1: Hello. Thank you for this comment. While 4 articles seems quite small, nearly 432/677 represent duplicate articles. Further, only 15 total articles discuss TTC outcomes in diabetic populations. However, we decided to only include articles that were comparative. This removed the final nine articles. We made this decision as all 15 studies report outcomes entirely differently from one another, including how union is defined. Including only comparative articles allows for more comparison, otherwise the review would still remain purely qualitative. Meaning, the additional nine articles would not add to the results besides simply summarizing their individual results.

Comment 2: It would be adding more value to have a forest plot to visualize some of the findings.

Response 2: Thank you for this suggestion. I completely agree that forest plots help visualize results, but they are not possible for this study. A forest plot would require that 2+ studies report the same metric. As metrics across all studies were slightly different, forcing the metrics to be interpreted the same in the setting of a forest plot would not be accurate. This lack of consistent reporting is frequently mentioned in the discussion in our suggestions for future research.

Comment 3: A more nuanced discussion on diabetes severity and its specific impact on outcomes could add more depth.

Response 3: This is a great point, thank you for this insightful comment. Please see lines 318-321, as I did present this comment as a possible limitation of this study. I have added an additional sentence to better elaborate on this point. Please see below for the text found in the manuscript.

“Second, this study strictly compared DM patients to non-diabetic patients. Limited studies have analyzed TTC arthrodesis results in patients with varying DM severity,[34] and a dedicated systematic review to this topic would greatly contribute to a comprehensive analysis of this procedure in DM patient populations. While diabetic severity, grouped as either “controlled-” or “uncontrolled-” diabetes in previous studies, did demonstrate higher TTC failure rates, [34] additional factors such as age, BMI, and length of diabetes diagnosis all may impact the success of this procedure. Therefore, there is a need for detailed future studies that helps understand the outcomes in DM patients while accounting for the previously mentioned confounding factors.”

Comment 4: The authors conclude that diabetic and non-diabetic patients have comparable functional outcomes, but can this be misleading? – given only few studies reviewed.

Response 4: I agree with you that it may be difficult to make any conclusive claims based on the limitations of this systematic review. However, this review represents the most comprehensive study to date, including 162 different patients. The limitations are clearly stated in the discussion and suggestions for future research are also outlined.

Comment 5: The study fails to establish a clear, standardized definition of success across studies. Is it functional measures? Or fusion rate?

Response 5: Thank you for this insightful comment. This was a large discussion point within this paper, as I believe the lack of standardized definition of success is a large limitation. Lines 281-290 discuss this comment in detail. A systematic review is incapable of coming up with a new definition of success, but rather, this review identifies the discrepancy in “success” metrics across the literature and makes calls for future research for certain metrics such as these.

Reviewer 3 Report

Comments and Suggestions for Authors

Thank you to the authors for completing this study and for their submission.

Please see and reply to all comments in the PDF attached.

Note, you need to review the PRISMA checklist and make sure every item of the checklist (https://www.prisma-statement.org/s/PRISMA_2020_expanded_checklist-rp3l.pdf) is addressed in the manuscript. There appears to be much missing.

Author Response

Reviewer 3:

We would like to thank you for your diligent review of our manuscript, both in terms of content and formatting. Please see below for responses to your comments.

Comment 1: From the checklist: If the purpose is to evaluate the effects of interven-tions, use the Population, Intervention, Comparator, Outcome (PICO) framework or one of its variants, to state the comparisons that will be made.

Response 1: After working with the editorial staff, the entire PRISMA checklist has been filled out and uploaded to the submission site.

Comment 2: In the text, reference numbers should be placed in square brackets [ ] and placed before the punctuation; for example [1], [1–3] or [1,3].

Response 2: I have edited this accordingly in the manuscript.

Comment 3: You have not established this abbreviation yet. ...diabetes mellitus (DM)-related...

Response 3: I have corrected this in the manuscript.

Comment 4: Was this study registered in PROSPERO (https://www.crd.york.ac.uk/prospero/)? If not, do so and include this in the manuscript. Also, several Methods checklist items are not addressed from the PRISMA checklist (https://www.prisma-state- ment.org/s/PRISMA_2020_expanded_checklist-rp3l.pdf).

Response 4: This manuscript was not pre-registered on PROSPERO. As MDPI only recommends and does not require registration, it was not performed. Further, as this review was performed nearly 18 months ago, registering the review now would not be intuitive. Future work from our team does adhere to MDPI’s registration guidelines.

Comment 5: Write out the abbreviation when it is first used followed by the abbreviation in parentheses.

Response 5: I have corrected this in the manuscript.

Comment 6: What were the tasks of the multiple authors? Were there protocols for disagreements between authors?

Response 6: I have added additional context in the manuscript on lines 97-98.

Comment 7: Make sure every item from the PRISMA checklist is addressed in the results.

Response 7: Same response as above.

Comment 8: This sentence does not make sense. It seems like it should be: A total of four articles met inclusion criteria out of the 677 articles yielded from the search terms.

Response 8: I have corrected this in the manuscript.

Comment 8-12: Footers for Table 2

Response 8-12: I have edited the table to include a footer with relevant abbreviations.

Comment 14: Write out abbreviation…

Response 14: I have edited this in the manuscript.

Comment 15: Convert to months for the purpose of easily comparing to the other studies.

Response 15: I have converted the requested metric to months.

Comment 16-19: Write out the abbreviation…

Response 16-19: I have corrected this in the manuscript.

Comment 20: Write out abbreviation in footer…

Response 20: I have edited this in the manuscript.

Comment 21: Abbreviation section edits

Response 21: Thank you for providing a full list of abbreviations that we did not include. I have updated this section accordingly.

Comment 22: Reference Section edits

Response 22: Hello, thank you for this comment. As we use a citation manager, manual adjustment of citation is difficult. Further, if DOI is not present, it was not listed on PubMed. In our past works with MDPI, the editorial office has been incredibly helpful with editing citations to match their requirements. I am happy to work with them to add any missing information, but the text is formatted according to the text style that is preset in the template.

Round 2

Reviewer 2 Report

Comments and Suggestions for Authors

Thank you for incorporating the feedback and updating the manuscript.